# Epidemiology of ankle sprains and anterior cruciate ligament injuries in youth basketball athletes in Niigata, Japan: A regional survey on injury management and healthcare professional involvement

Takanori Kikumoto[1]*, Jun Mizutani[2], Atsushi Tsuchida[3], Tomoya Takabayashi[1], Masayoshi Kubo[1], Hirokazu Okada[3], Yoshiaki Kon[2]

1 Department of Physical Therapy, Niigata University of Health and Welfare, Niigata, Japan, 2 Department of Physical Therapy, Kon Orthopedic Clinic, Niigata, Japan, 3 Department of Physical Therapy, Okada Orthopedic Clinic, Niigata, Japan

* kikumoto@nuhw.ac.jp

## Abstract

### Objective

Ankle sprains are highly prevalent among youth basketball players, particularly in the under-15 age group, and inadequate post-injury management may contribute to the development of chronic ankle instability and may be associated with an increased risk of anterior cruciate ligament injuries. This study aimed to determine the incidence of ankle sprains and post-injury management among under-12 and under-15 basketball players through a large-scale questionnaire survey. Additionally, we examined the prevalence of chronic ankle instability and anterior cruciate ligament injuries to assess the epidemiological impact of ankle sprains on young athletes. By examining these age groups, this study aimed to provide insights into potential preventive strategies and the need for appropriate medical intervention during early athletic development.

### Methods

A cross-sectional survey was conducted to assess ankle sprain history, medical consultation rates, prevalence of chronic ankle instability, and anterior cruciate ligament injuries. The study targeted school and club basketball teams in Niigata, Japan. Overall, 2,747 youth basketball players from under-12 and under-15 age groups participated.

### Results

Ankle sprains were more common in under-15 players (61.2%) than in under-12 players (39.0%). Medical consultation rates were lower in under-15 players (56.6%)

**Data availability statement:** All relevant data are within the manuscript and its Supporting Information files.

**Funding:** The author(s) received no specific funding for this work.

**Competing interests:** The authors have declared that no competing interests exist.

than in under-12 players (80.1%). Chronic ankle instability was found in 12.4% of the players. Among under-15 players, 0.9% underwent anterior cruciate ligament reconstruction, and half of them had chronic ankle instability. Many players returned to play despite experiencing pain.

## Conclusions

The coexistence of chronic ankle instability and anterior cruciate ligament injuries suggests a possible association between these conditions; however, causal relationships cannot be inferred because of the cross-sectional study design and the small number of anterior cruciate ligament cases.

## Introduction

Ankle sprains are among the most frequently occurring injuries [1] and are the most common injuries observed in basketball players [2–4]. Additionally, they have a high recurrence rate of 56–74% [5]. Repeated injuries can result in ligament reinjury and residual instability of the ankle joint, ultimately leading to chronic ankle instability (CAI) [6]. CAI is a precursor to ankle osteoarthritis, with some cases requiring surgical intervention [3]. Furthermore, ankle sprains have been identified as a risk factor for anterior cruciate ligament (ACL) injuries, which are the most severe knee injuries occurring in basketball [2]. Given these concerns, establishing effective methods to prevent ankle sprains is crucial for reducing ACL injury risk.

A history of ankle sprains is considered the most significant risk factor for future sprains. Basketball players with a history of ankle injuries had a fivefold higher likelihood of developing another ankle injury compared with those without past injuries [3]. Many other studies have also identified a history of ankle sprains as a major risk factor for recurrence [2,5,6]. One of the most effective ways to prevent recurrent sprains and chronic instability is to implement appropriate rehabilitation following the initial injury [7]. However, as mentioned earlier, both the recurrence rate of ankle sprains as well as the prevalence of CAI remains high among basketball players.

Despite the high incidence of ankle sprains in basketball, they may not be given sufficient attention in sports settings. Regarding the classification of ankle sprain severity, Grade 1 sprains are defined as slight stretching and minor damage to the ligament fibers. More severe injuries are likely to be reported to medical staff or receive medical attention, making them more apparent in injury statistics. However, Grade 1 sprains often allow for an immediate return to play, and in many cases, athletes do not seek medical consultation, potentially leading to underreporting in epidemiological data. Thus, athletes' injury histories, which are a key risk factor for recurrent ankle sprains, may not be accurately documented, and inadequate rehabilitation before returning to play creates a cycle of repeated injury.

For decades, a four-stage process for sports injury prevention has been proposed [8]. However, the first stage—epidemiological investigation—may not be accurately conducted. Supporting this concern, an epidemiological study has indicated that over

55% of athletes do not seek medical evaluation after sustaining an ankle sprain [3], highlighting the challenge of accurately assessing the true extent of the issue. Previous research has also reported that 90% of individuals who sustained an ankle sprain did not receive rehabilitation from a medical professional within 30 days of developing the injury [7].

In the present study, we specifically focused on under-12 (U-12) and under-15 (U-15) basketball players, because early prevention of ankle sprains and ACL injuries in younger athletes is crucial for reducing the risk of these injuries as they continue to develop their motor skills and sport-specific movements [9]. The physical growth and musculoskeletal changes that occur during these years may influence injury susceptibility, particularly in sports such as basketball, which involve dynamic, high-impact movements [10]. Furthermore, patterns of healthcare utilization and rehabilitation practices during these formative years could markedly impact future injury risk and long-term joint health [8]. Therefore, the present study aimed to investigate the actual incidence of ankle sprains and post-injury management among U-12 and U-15 basketball players through a large-scale questionnaire survey. Additionally, the study examined the prevalence of CAI and ACL injuries to assess the epidemiological impact of ankle sprains on young athletes. By examining these age groups, this study aimed to provide insights into potential preventive strategies and the need for appropriate medical intervention during early athletic development [8].

## Participants and methods

### Participants and study design

This study targeted a total of 9,914 basketball players from 566 teams who were affiliated with the Niigata Basketball Association and consented to participate. The U-12 category comprised 4,968 players from 277 teams, while the U-15 category included 4,946 players from 289 teams. The survey period extended from February 1 to April 30, 2020.

Niigata Prefecture, with a population of 2.4 million, is a major urban center in Japan and was selected as an optimal location for understanding the actual conditions examined in this study. Participants were informed of the study through a mobile application used by the Niigata Basketball Association, and paper-based questionnaires were distributed and collected via team coaches. For younger players, parents or guardians assisted with questionnaire completion, particularly for medical history items. This hybrid approach ensured accessibility and facilitated higher response rates. This study was designed as an observational (cross-sectional) study and was conducted according to the Strengthening the Reporting of Observational Studies in Epidemiology statement, which provides guidelines for reporting observational research. The study protocol was approved by the research ethics committee of Niigata University of Health and Welfare (approval number: 18583–210218). Written informed consent was obtained from the parents or legal guardians of all participating minors, and assent was obtained from the players themselves, in accordance with the Declaration of Helsinki.

### Assessment items

The study assessed the participants' history of ankle sprains, defined as an acute injury to the ankle joint caused by an inversion or eversion mechanism, resulting in pain, swelling, or functional limitation sufficient for the injury to be recognized by the player or coach.

In addition, the Cumberland Ankle Instability Tool (CAIT), which has demonstrated good test–retest reliability and internal consistency in previous validation studies, was used to identify CAI cases. Participants were classified as having CAI if they met the following five criteria on the same side: (1) their first ankle sprain occurred more than 1 year ago, (2) they had no history of ankle sprains within the previous 3 months, (3) they had experienced at least two ankle sprains on the same side, (4) they had experienced at least two episodes of ankle giving way within the previous 6 months, and (5) they had a Cumberland Ankle Instability Tool score of 24 or lower. Based on these criteria, the proportion of participants with unilateral or bilateral CAI was calculated. Additionally, the proportion of participants with at least one prior ankle sprain was determined and so was their subsequent management. The study also examined whether participants had undergone ACL reconstruction surgery owing to an ACL rupture. ACL reconstruction history was self- or parent-reported and

was not verified using medical records. All datasets were fully anonymized prior to analysis and public sharing. No personally identifiable information, including names, team identifiers, or contact details, was retained in the shared dataset.

For statistical analysis, we compared the proportions of outcomes between U-12 and U-15 players using chi-square ($\chi^2$) tests. For each main comparison, we calculated odds ratios (ORs) with 95% confidence intervals (95% CIs) to quantify effect sizes, treating the U-12 group as the reference. In the comparison of ACL reconstruction between age groups, where one cell contained zero events, we applied a continuity correction by adding 0.5 to each cell before calculating the OR and 95% CI. Statistical analyses were performed using R Statistical Software (R Foundation for Statistical Computing), with the significance level set at 5%.

## Results

Valid responses were obtained from 2,747 participants, including 1,400 U-12 players (735 male players, mean age: 9.84 ± 1.40 years; 665 female players, mean age: 9.81 ± 1.38 years) and 1,347 U-15 players (711 male players, mean age: 13.40 ± 0.94 years; 636 female players; mean age: 13.30 ± 0.92 years) (Table 1). The corresponding response rates were 28.2% (128 teams) and 27.2% (112 teams), and the overall response rate was 27.7%. The competition levels of the teams ranged widely, from national tournament-level teams to recreational-level teams. A history of ankle sprains was reported by 546 (39.0%) and 824 (61.2%) players in the U-12 and U-15 groups, respectively, with the prevalence being significantly higher in the latter (OR 2.46, 95% CI 2.11–2.87, $\chi^2$ = 135.0, df = 1, P < 0.001; Table 2A).

Although injury severity was subjectively assessed by the players or their coaches, contact with an orthopedic or other medical institution was reported by 1,134 (80.1%) and 762 (56.6%) players in the U-12 and U-15 groups, respectively, with the rate being significantly higher in the U-12 group (OR 0.31, 95% CI 0.26–0.36, $\chi^2$ = 191.6, df = 1, P < 0.001; Table 2B). Professional involvement was defined as consultation or treatment by licensed healthcare professionals, including orthopedic surgeons, physicians, physical therapists, or nationally licensed Judo therapists in Japan. Players who reported not seeking medical attention numbered 120 (8.6%) in the U-12 group and 354 (26.3%) in the U-15 group, with the rate being significantly lower in the former (P < 0.05). Additionally, players who sought care from a judo therapist (judo seifukushi), a nationally licensed practitioner in Japan certified by the Ministry of Health, Labor and Welfare who provides non-surgical management of acute musculoskeletal injuries such as sprains, contusions, and muscle strains; and who treats fractures and dislocations only with physician consent except for emergency first aid, included 158 (11.3%) and 230 (17.1%) in the U-12 and U-15 groups, respectively (Table 1). Regarding return to play, 766 players (54.7%) in the U-12 group and 721 players (53.5%) in the U-15 group resumed competition despite persistent pain in the injured ankle with no significant difference between age groups (OR 0.95, 95% CI 0.82–1.11, $\chi^2$ = 0.39, df = 1, P = 0.53; Table 2C). Based on the inclusion criteria of the International Ankle Consortium, 173 (12.4%) and 167 (12.4%) players in the U-12 and U-15 groups, respectively, were identified as having CAI, with similar prevalence between age groups (OR 1.00, 95% CI 0.80–1.26, $\chi^2$ = 0.00, df = 1, P = 0.97; Table 2D). Furthermore, no players in the U-12 group had undergone ACL reconstruction surgery, whereas 12 (0.9%) in the U-15 group did. Using a continuity correction because of the zero cell in the U-12 group, the odds of having a history of ACL reconstruction were substantially higher in U-15 than in U-12 players (OR 26.22, 95% CI 1.55–443.26, $\chi^2$ = 11.6, df = 1, P < 0.001), although the absolute number of ACL-reconstructed players was small. Among U-15 players who had undergone ACL reconstruction, six (50%) developed CAI in the same ankle (Table 2D).

**Table 1. Participant demographics.**

| Age group | Total participants | Male group | Female group | Age (Mean ± SD) |
|-----------|--------------------|-----------|--------------|-----------------|
| U-12 | 1,400 | 735 | 665 | M: Age (9.84 ± 1.40), F: Age (9.81 ± 1.38) |
| U-15 | 1,347 | 711 | 636 | M: Age (13.40 ± 0.94), F: Age (13.30 ± 0.92) |

**Table 2. Survey results.**

**Table 2A. Incidence of Ankle Sprains**

| Age group | With event (n, %) | Without event (n, %) | OR | 95% CI | χ² | P value | |
|---|---|---|---|---|---|---|---|
| U-12 | 546 (39.0%) | 854 (61.0%) | - | - | - | - | |
| U-15 | 824 (61.2%) | 523 (38.8%) | 2.46 | 2.11–2.87 | 135.0 | < 0.001 | |

**Table 2B. Healthcare Consultation**

| Age group | Yes (n, %) | No (n, %) | OR | 95% CI | χ² | P value | |
|---|---|---|---|---|---|---|---|
| U-12 | 1134 (80.1%) | 266 (19.9%) | - | - | - | - | |
| U-15 | 762 (56.6%) | 585 (43.4%) | 0.31 | 0.26–0.36 | 191.6 | < 0.001 | |

**Table 2C. Return to Play Despite Pain**

| Age group | Returned with pain (n, %) | No pain (n, %) | OR | 95% CI | χ² | P value | |
|---|---|---|---|---|---|---|---|
| U-12 | 766 (54.7%) | 634 (45.3%) | - | - | - | - | |
| U-15 | 721 (53.5%) | 626 (46.5%) | 0.95 | 0.82–1.11 | 0.39 | 0.53 | |

**Table 2D. Prevalence of CAI and ACL Reconstruction Surgery**

| Condition | Age group | With condition (n, %) | Without condition (n, %) | OR | 95% CI | χ² | P value |
|---|---|---|---|---|---|---|---|
| CAI | U-12 | 173 (12.4%) | 1227 (87.6%) | - | - | - | - |
| | U-15 | 167 (12.4%) | 1180 (87.6%) | 1.00 | 0.80–1.26 | 0.00 | 0.97 |
| ACL | U-12 | 0 (0.0%) | 1400 (100%) | - | - | - | - |
| | U-15 | 12 (0.9%) | 1335 (99.1%) | 26.22 | 1.55–443.26 | 11.6 | < 0.001 |

CAI: chronic ankle instability, ACL: anterior cruciate ligament

## Discussion

This study aimed to determine the actual incidence of ankle sprains and post-injury management among U-12 and U-15 basketball players through a large-scale questionnaire survey. The study also examines the prevalence of CAI and ACL injuries to assess the epidemiological impact of ankle sprains on young athletes. The results of this study reveal that the incidence of ankle sprain during basketball competition was significantly higher in the U-15 age group. Previous studies have also reported that the risk of ankle sprain increases with growth owing to increased competition intensity and game frequency [8,11]. However, given that a considerable number of ankle sprains also occur in the U-12 age group, early education on injury prevention strategies and promotion of higher medical consultation rates among younger athletes are important. Studies have shown that early intervention and appropriate management of ankle sprains can reduce the risk of recurrent injuries and long-term complications [3]. When comparing the incidence of ankle sprains across different sports, basketball, with its frequent directional changes and high-impact landings, presents a higher risk than that associated with many other sports, particularly sports with less physical contact and less frequent sharp movements, sports in which ankle sprains are generally less common. [12,13].

Additionally, the proportion of players who sought medical attention after an ankle sprain was significantly lower in the U-15 group than in the U-12 group. Prior research has indicated that as competition levels rise, athletes tend to rely more on self-assessment or coaches' judgment, leading to inadequate medical intervention [14,15]. Moreover, players with a history of CAI may have a higher rate of re-injury than those without, suggesting that CAI increases the likelihood of future sprains, further complicating recovery and rehabilitation [16,17]. Furthermore, CAI remained consistent at approximately 12.4% in both age groups. Some athletes with CAI also experienced ACL injuries, suggesting that CAI may increase the

risk of other lower limb injuries [17,18]. Although the CAI classification criteria used in this study were based on established definitions commonly applied in previous research, these criteria were originally developed for adult populations. Therefore, caution is warranted when interpreting CAI prevalence in younger athletes, particularly in the U-12 age group, because developmental differences in neuromuscular control, injury perception, and symptom reporting may influence questionnaire responses and classification outcomes. The coexistence of CAI and ACL reconstruction observed in some players suggests a possible association; however, causal relationships cannot be inferred due to the cross-sectional design and the small number of ACL cases. This highlights the importance of maintaining ankle stability in sports such as basketball, which involve rapid directional changes and landing movements. Previous studies have also indicated that CAI can lead to biomechanical abnormalities in the knee and hip joints, increasing the likelihood of more severe injuries [2,19].

Several epidemiological studies have reported varying prevalence rates of CAI and ACL injuries across different sports and age groups. For instance, previous research has indicated that the prevalence of CAI among young athletes ranges from 9% to 23%, depending on the sport and competition level [20]. In basketball, the frequent occurrence of ankle sprains and the high risk of re-injury contribute to a relatively high CAI prevalence compared with that in other sports such as soccer and volleyball [21]. Similarly, ACL injuries have been extensively studied in young athletes, with reports showing a higher incidence among female basketball players because of biomechanical and hormonal factors [22]. While this study finds a consistent CAI prevalence of approximately 12.4% in both age groups, other studies have reported slightly higher rates in older age groups, possibly owing to accumulated exposure to high-impact movements and inadequate rehabilitation following previous injuries [23]. These variations in CAI and ACL injury prevalence highlight the need for sport-specific and age-specific injury prevention strategies. Furthermore, a meta-analysis examining return-to-play outcomes following ACL reconstruction has shown that delayed or inadequate rehabilitation is a major factor contributing to poor functional outcomes and increased re-injury risk [24]. These findings emphasize the importance of structured rehabilitation programs and medical supervision to mitigate the long-term consequences of ankle and knee injuries.

The findings of this study suggest that a history of ankle sprains not only increases the risk of recurrence during sports activities but also may serve as a risk factor for more severe injuries, such as ACL tears [25]. Notably, the lower rate of medical consultation after injury in the U-15 group may lead to inadequate rehabilitation and treatment, potentially contributing to a higher incidence of CAI [26]. Research has shown that insufficient rehabilitation after an initial ankle sprain increases the likelihood of developing CAI [27].

Moreover, this study finds that approximately half of the players returned to competition while still experiencing residual pain in the injured ankle. Continuing to play despite persistent pain can lead to impaired ankle function and an increased risk of re-injury. Existing research has also indicated that returning to competition with pain can cause a decline in athletic performance and biomechanical abnormalities, ultimately increasing the risk of re-injury [28,29]. A significant issue highlighted in this study is that the severity of ankle sprains is often determined by self-assessment or coaches' judgment. In sports settings, minor sprains are frequently overlooked, leading to inadequate rehabilitation and ultimately increasing the risk of developing CAI or sustaining more severe injuries [17,30].

These findings underscore the importance of proper management and rehabilitation of ankle sprains to prevent CAI and ACL injuries. For U-15 athletes, education and guidance are essential to ensure proper evaluation and treatment after injury. Additionally, strengthening the involvement of medical staff in sports settings and ensuring appropriate care even for minor sprains are necessary to reduce the risk of re-injury [31]. Previous research has also shown that rehabilitation programs supervised by healthcare professionals lower the risk of developing CAI, which aligns with the results of this study [32].

## Limitations

The study was limited to one prefecture, Niigata, in Japan, which may affect the generalizability of the results to wider populations. In addition, this study relied on retrospective self- or parent-reported questionnaire data. Therefore, recall

bias cannot be ruled out, particularly for mild ankle sprains that may have been forgotten or misclassified by respondents. However, while Niigata has unique demographic and geographic features, these are unlikely to significantly affect the epidemiology of ankle sprains and CAI in youth basketball players, and the relatively large sample size and diverse demographic composition allows the results to be interpreted as reflective of broader trends in Japanese youth basketball players. Moreover, even if regional characteristics do influence the findings, identifying these factors provides valuable information for other regions, emphasizing the importance of broader nationwide or international studies to confirm these results. Because the overall response rate was 27.7%, non-response bias cannot be excluded. We were unable to compare responders and non-responders in detail, and therefore the findings may not fully represent all youth basketball players in the region.

In addition, the CAI classification criteria used in this study were originally developed primarily for adult populations. Although these criteria enabled standardized classification, developmental differences in younger athletes, particularly in the U-12 group, may influence symptom perception and questionnaire responses. Therefore, CAI prevalence estimates in younger players should be interpreted with caution and considered descriptive rather than diagnostic.

The results of this study indicate that the incidence of ankle sprains in basketball was high, particularly in the U-15 age group. Furthermore, the low rate of medical consultation after injury and inadequate rehabilitation may have contributed to an increased risk of CAI and ACL injuries. Notably, many athletes returned to competition while still experiencing pain, which may have created a cycle of recurrent injuries. Given that ankle sprains also occurred frequently in the U-12 age group, it is crucial to emphasize injury prevention and proper management from an early age. Studies have suggested that early intervention and appropriate rehabilitation can reduce the likelihood of chronic complications and improve long-term athletic performance. The relatively young age of some participants, particularly in the U-12 group, may limit their ability to fully meet all CAI criteria. Therefore, CAI prevalence in younger players should be interpreted descriptively rather than diagnostically.

## Clinical implications

The findings of this study provide important insights into the management of ankle sprains in basketball and could serve as a fundamental knowledge base for establishing future preventive measures. Although the coexistence of CAI and ACL reconstruction observed in some players suggests a possible association, causal relationships cannot be inferred because of the cross-sectional design of this study and the small number of ACL cases. Nevertheless, future nationwide or international studies are needed to confirm these findings and develop comprehensive injury prevention strategies for young athletes. Therefore, in addition to ensuring proper management and rehabilitation of ankle sprains, strengthening the involvement of medical professionals in sports settings from a young age is essential to mitigate the risks associated with repeated injuries.

## Supporting information

**S1 Dataset. Fully anonymized dataset of questionnaire responses used for the statistical analyses.**
(XLSX)

**S2 Checklist. STROBE checklist for cross-sectional studies.**
(DOCX)

## Author contributions

**Conceptualization:** Takanori Kikumoto, Yoshiaki Kon.

**Data curation:** Takanori Kikumoto, Jun Mizutani, Atsushi Tsuchida.

**Formal analysis:** Takanori Kikumoto, Jun Mizutani, Atsushi Tsuchida.

**Investigation:** Takanori Kikumoto, Jun Mizutani, Atsushi Tsuchida.

**Methodology:** Takanori Kikumoto.

**Project administration:** Takanori Kikumoto, Jun Mizutani, Yoshiaki Kon.

**Software:** Takanori Kikumoto.

**Supervision:** Takanori Kikumoto, Masayoshi Kubo.

**Validation:** Takanori Kikumoto, Tomoya Takabayashi, Masayoshi Kubo, Hirokazu Okada, Yoshiaki Kon.

**Visualization:** Takanori Kikumoto, Jun Mizutani, Atsushi Tsuchida.

**Writing – original draft:** Takanori Kikumoto.

**Writing – review & editing:** Takanori Kikumoto, Tomoya Takabayashi, Masayoshi Kubo, Hirokazu Okada, Yoshiaki Kon.

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
