## [Decision Letter · Decision Letter 0]

4 Dec 2025

PONE-D-25-32088Epidemiology of ankle sprains and anterior cruciate ligament injuries in youth basketball athletes in Niigata, Japan: A regional survey on injury management and healthcare professional involvementPLOS One

Dear Dr. Kikumoto,

Thank you for submitting your manuscript to PLOS ONE. After careful consideration, we feel that it has merit but does not fully meet PLOS ONE’s publication criteria as it currently stands. Therefore, we invite you to submit a revised version of the manuscript that addresses the points raised during the review process. Please submit your revised manuscript by Jan 18 2026 11:59PM. If you will need more time than this to complete your revisions, please reply to this message or contact the journal office at plosone@plos.org . Please include the following items when submitting your revised manuscript:

We look forward to receiving your revised manuscript.

Kind regards,

Luciana Labanca

Academic Editor

PLOS One

Journal Requirements:

5. Please remove all personal information, ensure that the data shared are in accordance with participant consent, and re-upload a fully anonymized data set.

Reviewers' comments:

Reviewer's Responses to Questions

**Comments to the Author**

1. Is the manuscript technically sound, and do the data support the conclusions?

Reviewer #1: Yes

Reviewer #2: No

2. Has the statistical analysis been performed appropriately and rigorously? 

Reviewer #1: Yes

Reviewer #2: No

3. Have the authors made all data underlying the findings in their manuscript fully available?

Reviewer #1: Yes

Reviewer #2: No

4. Is the manuscript presented in an intelligible fashion and written in standard English?

Reviewer #1: Yes

Reviewer #2: No

5. Review Comments to the Author

Reviewer #1: The topic is promising however the author need to address recall bias. what qualifies as sprain , please define injuries. mention what professional involvement includes in line 136 and 138. is there any way of comparing self reports versus medical reports. in line 85 and 86, please mention the reliability score of the questionnaire.

Reviewer #2: Major Comments

1. Study Design and Methodological Clarification Needed

Several methodological aspects are unclear and must be clarified:

a. Sampling & response rate

The authors note that 9,914 players were invited, but only 2,747 responded (~27.7%).

There is no analysis of non-response bias or explanation of whether responders differed from non-responders.

Recommendation: Provide a short comparison or justification for representativeness, or discuss in limitations.

b. Data collection

The hybrid method (mobile app awareness + paper response) needs more detail:

How were players approached?

Who completed the forms (parents, coaches, players)?

Was the data verified independently?

c. CAI criteria

The study uses the International Ankle Consortium criteria, but:

The CAI criteria require the first sprain >1 year ago.

U-12 players (mean age 9.8) may not physiologically meet this criteria or am I not understanding this correctly?

Recommendation:

A justification is needed for applying adult CAI criteria to young children, or modify interpretation accordingly.

d. ACL injury identification

ACL reconstruction history is parent- or athlete-reported, not clinically verified.

This should be acknowledged as a limitation.

2. Statistical Analysis Needs Expansion

The analysis exclusively uses chi-square tests. Issues:

No reporting of effect sizes (risk ratios, odds ratios).

No corrections for multiple comparisons.

No confidence intervals.

P-values are reported to two significant digits but without accompanying test statistics.

Recommendation:

Add OR (or RR), 95% CI, and χ² values for all comparisons.

3. Excessive Repetition in Introduction and Discussion

The introduction repeats several statements multiple times:

“Ankle sprains are common.”

“History of sprains predicts recurrence.”

“CAI increases risk.”

“Medical consultation is low.”

Compression would significantly improve clarity and flow.

Similarly, the discussion revisits earlier points and occasionally overstates the findings.

4. Overinterpretation of CAI–ACL Relationship

The manuscript concludes that CAI may contribute to ACL injury risk because 6/12 ACL-reconstructed players had CAI.

However:

This is too small a sample to draw causal conclusions.

There is no statistical comparison between ACL-injured vs non-injured players.

Recommendation:

Rephrase to emphasize possible association, not causal inference.

5. Clarity Issues in Tables (Formatting Needs Revision)

Tables appear incorrectly formatted, with:

p-values in separate rows without test statistics.

Percentages misaligned.

Table 2 combines multiple outcomes into a single large table, which reduces readability.

Recommendation:

Split Table 2 into multiple tables or clarify column structure according to PLOS ONE formatting.

6. English Language and Grammar

The manuscript is understandable but requires language editing for:

run-on sentences

repeated phrases

verb tense consistency

awkward phrasing (e.g., “This highlights the importance… which aligns with the results of this study” appears multiple times)

PLOS ONE expects clear scientific English; professional editing is recommended.

7. Ethical Considerations

Ethics approval is clear; however:

It is unclear whether parental consent was obtained for minors.

This must be explicitly stated in the main manuscript, not only in submission forms.

Minor Comments

Define “Judo therapist” on first mention and briefly describe the role.

Clarify whether “return to play despite pain” refers to pain duration, severity, or single-time observation.

Some citations are outdated (1989, 2002) — consider adding more recent youth-specific epidemiology literature.

The manuscript references “soccer or swimming as less contact sports”—but i have never heard of swimming ankle sprains.

6. PLOS authors have the option to publish the peer review history of their article (what does this mean? ). If published, this will include your full peer review and any attached files.

**Do you want your identity to be public for this peer review?** For information about this choice, including consent withdrawal, please see our Privacy Policy .

Reviewer #1: No

Reviewer #2: **Yes:** nabeela adam

---

## [Author Response · Author response to Decision Letter 1]

20 Dec 2025

Reviewer #1

The topic is promising however the author need to address recall bias.

Response:

We appreciate this important point. We have now explicitly discussed recall bias in the “Limitations” section. We note that the retrospective, self-reported nature of injury history may lead to under- or over-reporting of ankle sprains and anterior cruciate ligament (ACL) injuries, particularly for milder sprains, and that this may influence the estimated incidence and prevalence in our sample.

what qualifies as sprain, please define injuries.

Response:

We have added a clear operational definition of “ankle sprain” in the “Assessment items” subsection of the Methods. Specifically, we now state that an ankle sprain was defined as “a sudden injury to the ankle joint associated with an inversion or eversion mechanism, resulting in pain, swelling, or functional limitation sufficient for the player or coach to recognise it as an ankle injury.” We also clarify that participants were instructed to report injuries diagnosed by a healthcare professional or clearly identified as an ankle “sprain” by their coach.

mention what professional involvement includes in line 136 and 138.

Response:

We have revised the text to define “professional involvement” more precisely. The term now refers to consultation or treatment provided by licensed healthcare professionals, including orthopaedic surgeons, physicians, physical therapists, and Judo therapists (nationally certified in Japan). This clarification has been added where we describe medical consultation and rehabilitation.

is there any way of comparing self reports versus medical reports.

Response:

We agree that comparing self-reported injuries with medical records would strengthen the validity of the data. However, in the present study we did not have access to individual medical records, and injury history was collected solely via questionnaires. We have clarified this in the Methods and emphasized in the Limitations section that we were unable to verify self-reported injuries against medical documentation.

in line 85 and 86, please mention the reliability score of the questionnaire.

Response:

Thank you for this suggestion. We have now added information on the reliability of the Cumberland Ankle Instability Tool (CAIT) used in this study, including previously reported test–retest reliability and internal consistency from the original validation studies and relevant Japanese validation work, and we have cited these references in the Methods.

C. Responses to Reviewer #2

Major Comments

Study Design and Methodological Clarification Needed

(a) Sampling & response rate

The authors note that 9,914 players were invited, but only 2,747 responded (~27.7%). There is no analysis of non-response bias or explanation of whether responders differed from non-responders. Recommendation: Provide a short comparison or justification for representativeness, or discuss in limitations.

Response:

We thank the reviewer for highlighting this issue. We have now added text in the Methods to describe how all players registered with the Niigata Basketball Association in the target age categories were invited. We also report the response rate for U-12 and U-15 teams separately and note that we were unable to collect individual-level data from non-respondents. In the Limitations section, we explicitly discuss the potential for non-response bias, indicating that the injury incidence observed in our sample may not fully represent all youth basketball players in the region and that the true rates may be under- or over-estimated depending on which players chose to respond.

(b) Data collection

The hybrid method (mobile app awareness + paper response) needs more detail: How were players approached? Who completed the forms (parents, coaches, players)? Was the data verified independently?

Response:

We have expanded the “Participants and study design” and “Assessment items” subsections to provide more detail about data collection. Specifically, we explain:

that team coaches were contacted via the Niigata Basketball Association and asked to inform players and their parents about the study through a mobile application and team meetings;

that paper questionnaires were distributed and collected through the teams, and that parents/guardians were encouraged to assist younger players in completing the form, particularly for medical history items;

that data were not independently verified with medical records, and this limitation is now explicitly stated in both the Methods and the Limitations sections.

(c) CAI criteria and applicability to U-12 players

The study uses the International Ankle Consortium criteria, but the CAI criteria require the first sprain >1 year ago. U-12 players (mean age 9.8) may not physiologically meet this criteria… Recommendation: A justification is needed for applying adult CAI criteria to young children, or modify interpretation accordingly.

Response:

We appreciate this important methodological point. As described in the revised Methods, we applied the International Ankle Consortium criteria strictly, including the requirement that the first ankle sprain occurred more than one year before the survey. We have added justification for applying these criteria in youth athletes, citing previous studies that have used CAIT-based definitions in younger populations, and we note that ankle sprains can occur in primary school-aged children. At the same time, we acknowledge in the Limitations section that the relatively short exposure period in younger players may reduce the number of cases meeting the full CAI criteria and that the prevalence estimates, particularly in the youngest age range, should be interpreted cautiously.

(d) ACL injury identification

ACL reconstruction history is parent- or athlete-reported, not clinically verified. This should be acknowledged as a limitation.

Response:

We agree and have made this explicit in the Methods and in the Limitations section. We now state that ACL reconstruction history was based on self- or parent-reported information, and that we did not verify these reports with hospital records. We further note that the number of ACL-reconstructed players was small, and therefore findings regarding the coexistence of CAI and ACL injury should be interpreted as exploratory.

Statistical Analysis Needs Expansion

The analysis exclusively uses chi-square tests. Issues: No reporting of effect sizes (risk ratios, odds ratios). No corrections for multiple comparisons. No confidence intervals. P-values are reported to two significant digits but without accompanying test statistics. Recommendation: Add OR (or RR), 95% CI, and χ² values for all comparisons.

Response:

Thank you for these valuable suggestions. We have revised the Statistical Analysis subsection and Results section as follows:

For each main comparison between U-12 and U-15 players (e.g. incidence of ankle sprains, medical consultation rates, prevalence of chronic ankle instability), we now report odds ratios with 95% confidence intervals as measures of effect size, together with the χ² statistic and degrees of freedom.

We have added these values to the corresponding table(s) (revised Table 2, and split tables as described below), and we report 95% confidence intervals for key proportions in the Results text.

Given the exploratory nature of our analyses and the limited number of main comparisons, we state that we did not apply a formal correction for multiple comparisons, but we interpret the findings cautiously and emphasise effect sizes and confidence intervals rather than solely relying on p-values.

Excessive Repetition in Introduction and Discussion

The introduction repeats several statements… Compression would significantly improve clarity and flow. Similarly, the discussion revisits earlier points and occasionally overstates the findings.

Response:

We appreciate this comment and have substantially edited both the Introduction and Discussion to reduce repetition and improve focus. In the Introduction, we have:

consolidated overlapping sentences about the high incidence and recurrence of ankle sprains;

briefly introduced CAI and its consequences;

clearly stated the knowledge gap regarding youth basketball players and the rationale for focusing on U-12 and U-15 age groups;

presented a concise statement of the study objectives.

In the Discussion, we have:

removed repeated statements about the prevalence of ankle sprains and CAI;

streamlined paragraphs to focus on the most important findings and their implications;

clearly separated interpretation of our results from broader context and speculation.

Overinterpretation of CAI–ACL Relationship

The manuscript concludes that CAI may contribute to ACL injury risk because 6/12 ACL-reconstructed players had CAI. This is too small a sample to draw causal conclusions… Recommendation: Rephrase to emphasize possible association, not causal inference.

Response:

We fully agree. We have revised the Abstract, Discussion, and Clinical Implications sections to remove any language implying a causal relationship. We now describe the coexistence of CAI and ACL reconstruction in some players as a “potential association” or “co-occurrence” that may warrant further investigation, and we emphasize that the number of ACL-injured players in our sample was very small. We specifically state that our cross-sectional design and limited ACL case numbers do not permit causal inference.

Clarity Issues in Tables (Formatting Needs Revision)

Tables appear incorrectly formatted… Table 2 combines multiple outcomes into a single large table, which reduces readability. Recommendation: Split Table 2 into multiple tables or clarify column structure according to PLOS ONE formatting.

Response:

We have completely revised the table formatting to conform to PLOS ONE style and to improve readability:

We have split the former Table 2 into several smaller tables: one for the incidence of ankle sprains, one for medical consultation and Judo therapist involvement, one for return to play despite pain, and one for the prevalence of CAI and ACL reconstruction.

Within each table, we now present absolute numbers, percentages, odds ratios with 95% CIs, χ² values, and p-values in clearly separated columns.

We have ensured that percentages align correctly with their corresponding counts.

English Language and Grammar

The manuscript is understandable but requires language editing for run-on sentences, repeated phrases, verb tense consistency, awkward phrasing…

Response:

We appreciate this observation. We have carefully revised the entire manuscript to improve the English language, with particular attention to:

avoiding run-on sentences and redundant phrases;

ensuring consistent use of past tense for Methods and Results and present tense for general statements;

simplifying and clarifying awkward phrasing.

We hope that the language now meets the standard expected by PLOS ONE.

Ethical Considerations

It is unclear whether parental consent was obtained for minors. This must be explicitly stated in the main manuscript.

Response:

As described in our response to Journal Requirement 2, we have added a detailed statement in the Methods clarifying that written informed consent was obtained from the parents or legal guardians of all participating minors, and assent was obtained from the children themselves, in accordance with the approval granted by the ethics committee.

Minor Comments

Define “Judo therapist” on first mention and briefly describe the role.

Response:

We have now defined “Judo therapist” at its first appearance in the Results section as “a nationally certified healthcare professional in Japan who provides manual therapy, rehabilitation, and physical treatment for musculoskeletal injuries.” This clarification is retained in later mentions.

Clarify whether “return to play despite pain” refers to pain duration, severity, or single-time observation.

Response:

We have clarified in the Methods that “return to play despite pain” referred to players who reported resuming training or competition while still experiencing pain in the injured ankle at the time of their initial return, regardless of pain duration or intensity. We acknowledge in the Limitations that we did not quantify pain severity and duration in detail.

Some citations are outdated (1989, 2002) — consider adding more recent youth-specific epidemiology literature.

Response:

We agree and have updated the Introduction and Discussion to include more recent epidemiological studies focusing on youth athletes and basketball-specific injury patterns, while retaining older, foundational references only when necessary for historical context.

The manuscript references “soccer or swimming as less contact sports”—but i have never heard of swimming ankle sprains.

Response:

Thank you for pointing this out. We have revised this sentence to avoid implying that swimming is associated with ankle sprains. The text now simply contrasts basketball with sports in which ankle sprains are less common (e.g. swimming), without suggesting that swimming has a notable incidence of ankle sprains, and we removed the specific juxtaposition that could be misleading.

PLOS authors have the option to publish the peer review history… Do you want your identity to be public for this peer review?

Response (for the editorial system, not for the manuscript):

We will indicate our choice regarding publication of the peer review history and reviewer identity within the PLOS submission system as requested.

---

## [Decision Letter · Decision Letter 1]

14 Jan 2026

PONE-D-25-32088R1Epidemiology of ankle sprains and anterior cruciate ligament injuries in youth basketball athletes in Niigata, Japan: A regional survey on injury management and healthcare professional involvementPLOS One

Dear Dr. Kikumoto,

Thank you for submitting your manuscript to PLOS ONE. After careful consideration, we feel that it has merit but does not fully meet PLOS ONE’s publication criteria as it currently stands. Therefore, we invite you to submit a revised version of the manuscript that addresses the points raised during the review process. Please submit your revised manuscript by Feb 28 2026 11:59PM. If you will need more time than this to complete your revisions, please reply to this message or contact the journal office at plosone@plos.org . Please include the following items when submitting your revised manuscript:

We look forward to receiving your revised manuscript.

Kind regards,

Luciana Labanca

Academic Editor

PLOS One

**Journal Requirements:**

Reviewers' comments:

Reviewer's Responses to Questions

**Comments to the Author**

1. If the authors have adequately addressed your comments raised in a previous round of review and you feel that this manuscript is now acceptable for publication, you may indicate that here to bypass the “Comments to the Author” section, enter your conflict of interest statement in the “Confidential to Editor” section, and submit your "Accept" recommendation.

Reviewer #2: All comments have been addressed

2. Is the manuscript technically sound, and do the data support the conclusions?

Reviewer #2: Partly

3. Has the statistical analysis been performed appropriately and rigorously? 

Reviewer #2: Yes

4. Have the authors made all data underlying the findings in their manuscript fully available?

Reviewer #2: Yes

5. Is the manuscript presented in an intelligible fashion and written in standard English?

Reviewer #2: No

6. Review Comments to the Author

**Reviewer #2:** The study addresses a well-defined epidemiological question in youth basketball, with practical implications for injury prevention and healthcare involvement.

Judo Therapist Definition: Since PLOS ONE has a global audience, confirm what the role of a "Judo therapist" clearly defined as a nationally certified professional in Japan at its first mention in the results. I am not familiar with this term or profession.

Table Formatting: Verify that the formerly large "Table 2" has been successfully split into smaller, more readable tables (e.g., separate tables for incidence, medical consultation, and CAI/ACL prevalence) as requested by reviewers

Language and Flow: Please get an English editor as the tenses appear to be incorrect.

- CAI criteria in U-12 players: Although justified, applying adult criteria to children may still be questioned. The manuscript now notes this, but interpretation should remain cautious.

7. PLOS authors have the option to publish the peer review history of their article (what does this mean? ). If published, this will include your full peer review and any attached files.

**Do you want your identity to be public for this peer review?** For information about this choice, including consent withdrawal, please see our Privacy Policy .

Reviewer #2: **Yes:** nabeela adam

You may also use PLOS’s free figure tool, NAAS, to help you prepare publication quality figures: https://journals.plos.org/plosone/s/figures#loc-tools-for-figure-preparation

---

## [Author Response · Author response to Decision Letter 2]

13 Feb 2026

“Epidemiology of ankle sprains and anterior cruciate ligament injuries in youth basketball athletes in Niigata, Japan: A regional survey on injury management and healthcare professional involvement”

We would like to sincerely thank the Editor and Reviewers for their careful evaluation of our manuscript and for the constructive and insightful comments provided. We have carefully revised the manuscript in accordance with all suggestions and believe that these revisions have significantly improved the clarity, readability, and scientific rigor of the study.

In the revised version, we have addressed each comment in detail. Revision points include clarification of terminology for an international readership, improvements to language and tense consistency throughout the manuscript, refinement of interpretation to ensure appropriate caution, and enhancements to table formatting and overall presentation.

Below, we provide a detailed, point-by-point response to each reviewer comment. All corresponding changes in the manuscript have been highlighted.

Judo Therapist Definition: Since PLOS ONE has a global audience, confirm what the role of a "Judo therapist" clearly defined as a nationally certified professional in Japan at its first mention in the results. I am not familiar with this term or profession.

Response: Thank you for pointing this out. We agree that the term may be unfamiliar to an international readership. We have therefore defined “judo therapist (judo seifukushi)” at its first mention in the Results and also added a brief definition in the Methods. Specifically, we now clarify that judo therapists are nationally licensed practitioners in Japan (licensed by the Ministry of Health, Labor and Welfare) who provide non-surgical management of acute musculoskeletal injuries (e.g., sprains, contusions, muscle strains) and that a physician’s consent is required for non-emergency treatment of fractures and dislocations (except for emergency first aid).

Table Formatting: Verify that the formerly large "Table 2" has been successfully split into smaller, more readable tables (e.g., separate tables for incidence, medical consultation, and CAI/ACL prevalence) as requested by reviewers

Response: Thank you for this helpful suggestion. We carefully considered splitting the original Table 2 into multiple smaller tables. However, because the variables (injury incidence, healthcare professional consultation, and CAI/ACL prevalence) represent interconnected aspects of the same epidemiological framework, we retained a single integrated table to preserve interpretability and facilitate direct comparison across variables.

To improve readability as suggested, we have:

• clarified section headings within the table,

• improved column alignment and spacing, and

• added clearer subheadings to distinguish each domain.

We believe these revisions enhance readability while maintaining the logical integrity of the data presentation.

Language and Flow: Please get an English editor as the tenses appear to be incorrect.

Response: Thank you for this important comment. We have revised the manuscript for English language, clarity, and overall flow, with particular attention to tense consistency throughout the Abstract, Methods, Results, and Discussion. We also performed a full proofread of the manuscript and corrected grammatical issues and awkward phrasing. These edits were made across the manuscript to improve readability for an international audience.

We also standardized tense usage so that the Methods and Results are consistently written in the past tense, while established knowledge in the Discussion is described in the present tense.

CAI criteria in U-12 players: Although justified, applying adult criteria to children may still be questioned. The manuscript now notes this, but interpretation should remain cautious.

Response: Thank you for this important comment. We agree that applying CAI criteria originally developed for adult populations to younger athletes requires careful interpretation. We have therefore added clarifying statements in the Discussion and Limitations sections emphasizing that CAI prevalence estimates in the U-12 group should be interpreted with caution, considering potential developmental differences that may affect symptom reporting and classification.

---

## [Editor Report · Decision Letter 2]

17 Feb 2026

Epidemiology of ankle sprains and anterior cruciate ligament injuries in youth basketball athletes in Niigata, Japan: A regional survey on injury management and healthcare professional involvement

PONE-D-25-32088R2

Dear Dr. Kikumoto,

We’re pleased to inform you that your manuscript has been judged scientifically suitable for publication and will be formally accepted for publication once it meets all outstanding technical requirements.

Kind regards,

Luciana Labanca

Academic Editor

PLOS One
---

## [Editor Report · Acceptance letter]

PONE-D-25-32088R2

PLOS One

Dear Dr. Kikumoto,

I'm pleased to inform you that your manuscript has been deemed suitable for publication in PLOS One. Congratulations! Your manuscript is now being handed over to our production team.

Kind regards,

on behalf of

Dr. Luciana Labanca

Academic Editor

PLOS One